# The Genomic Variant Store: Decoupling Cloud-Native Storage from Analysis to Power Population-Scale Genomic Services

### M. Morgan Aster
Broad Institute of MIT and Harvard
Cambridge, MA, USA
morgan.aster@broadinstitute.org

### M. Kate Balaconis
Broad Institute of MIT and Harvard
Cambridge, MA, USA
kate.balaconis@broadinstitute.org

### Matt Bemis
Broad Institute of MIT and Harvard
Cambridge, MA, USA
mbemis@broadinstitute.org

### Miguel Covarrubias
Broad Institute of MIT and Harvard
Cambridge, MA, USA
mcovarrubias@broadinstitute.org

### Aaron Hatcher
Broad Institute of MIT and Harvard
Cambridge, MA, USA
ahatcher@broadinstitute.org

### Christopher Kachulis
Broad Institute of MIT and Harvard
Cambridge, MA, USA
ckachulis@broadinstitute.org

### Sofia Labrecque
Broad Institute of MIT and Harvard
Cambridge, MA, USA
slabrecq@broadinstitute.org

### Saloni P. Shah
Broad Institute of MIT and Harvard
Cambridge, MA, USA
saloni.shah@broadinstitute.org

### Lee Lichtenstein
Broad Institute of MIT and Harvard
Cambridge, MA, USA
leel@broadinstitute.org

## ABSTRACT

The *All of Us* Research Program requires genomic infrastructure that scales to hundreds of thousands of samples without full reprocessing. We present the Genomic Variant Store (GVS), a cloud-native variant database on Google BigQuery deployed across 535,662 whole-genome sequencing samples in the *All of Us* CDR v9. By decoupling storage from analysis formats (persisting sparse alt-allele data that can be extracted to any downstream representation on demand), GVS functions as a platform for a growing suite of data products and applications. The *All of Us* Cohort Builder leverages GVS to enable researchers to identify all participants carrying any variants of interest across 1.3 billion sites, including intersecting that with phenotype data, and render a whole genome VCF for about USD\$80 (*N*=5,000 samples of 414,000 in CDR v8). The *All of Us* + AnVIL Imputation Service builds on GVS to deliver cloud-native array and low-pass whole genome sequencing (lpWGS) backed by the world's largest, most diverse reference panel of >515,000 whole genomes. Together, these demonstrate that cloud-native, extensible variant storage is foundational infrastructure for population-scale biomedical discovery.

**VLDB Workshop Reference Format:**
M. Morgan Aster, M. Kate Balaconis, Matt Bemis, Miguel Covarrubias, Aaron Hatcher, Christopher Kachulis, Sofia Labrecque, Saloni P. Shah, and Lee Lichtenstein. The Genomic Variant Store. VLDB 2026 Workshop: Biomedical Data Management Systems (BioDMS).

**VLDB Workshop Artifact Availability:**
The source code, data, and/or other artifacts have been made available at https://github.com/broadinstitute/gatk/tree/ah_var_store.

## 1 THE CHALLENGE: JOINT CALLING AT BIOBANK SCALE

Joint variant calling—simultaneously genotyping all samples using combined read evidence—produces a unified callset that improves genotype accuracy, particularly for rare variants where additional samples provide statistical context for distinguishing true low-frequency alleles from sequencing noise. The precision gain over single-sample merging is meaningful but not dramatic; the greater value of joint calling at biobank scale is the ability to consolidate data from hundreds of thousands of individuals into a single, consistent, queryable callset at very low marginal cost per sample. Analyses that would be impractical with fragmented, per-sample files (rare-variant association studies, population-level imputation, real-time cohort queries) become tractable with a unified callset at this scale.

Achieving this at scale introduces two compounding infrastructure problems. First, naive storage of the full genotype matrix grows quadratically with sample count. Second, the "N+k problem": when $k$ new samples arrive, re-running joint calling on all $N+k$ samples wastes prior compute. For a continuously-enrolling cohort like *All of Us*, full reprocessing is unsustainable. Figure 1 showcases GVS's process of ingestion, training, and producing analysis-ready data formats.

## 2 THE GENOMIC VARIANT STORE

GVS is a cloud-native variant database developed at the Broad Institute addressing scaling problems via sparse data modeling, incremental ingestion, and a strict separation between storage and analysis variant representations. GVS is an implementation of the Scalable Variant Call Representation [6]. This sparse representation causes storage to scale linearly with cohort size; by contrast, VCF encodes every position for every sample, so storage grows superlinearly as both sample count and the number of discovered variant sites increase.

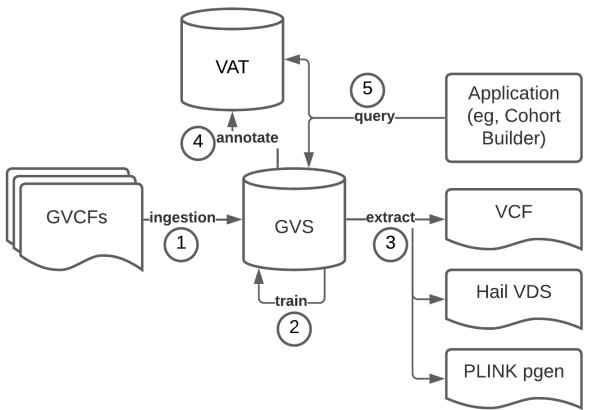

**Figure 1: GVS process for loading samples through joint calling and application support. (1) GVCFs are ingested into a BigQuery database with a specialized schema for variants at scale. (2) A filtering model (VETS) is trained to increase precision. (3) Samples and variants can be extracted into an analysis-ready format for consumption by researchers; GVS supports filtering by genomic region and/or sample set to reduce costs. (4) The Variant Annotation Table (VAT) can be rendered, including standard functional annotations (gene, protein change, variant classification) and allele frequency information from GVS and gnomAD. (5) GUI-based applications can query the GVS table for variant information, joinable with the VAT for more powerful queries. Operations and troubleshooting staff can use GVS tables to quickly locate data errors (not shown).**

## 2.1 Architecture and Data Model

GVS is built on Google BigQuery, a serverless columnar data warehouse. Because most positions in any genome are reference-homozygous, GVS stores only positions where at least one sample carries an alternate allele. Table 1 describes the key tables.

**Table 1: Simplified GVS BigQuery table schema.**

| Table | Description |
| --- | --- |
| sample_info | Maps sample identifiers to internal GVS IDs; stores per-sample metadata (sex, batch). |
| ref_ranges_* | Reference-block intervals for positions where all samples are homozygous-reference. Sharded by sample. |
| vet_* | Variant evidence table: one row per sample per alt-allele site (allele depth, genotype likelihoods, allele-specific annotations). Sharded by genomic position. |
| filter_set_* | VETS filter labels per variant site, referencing the model trained on the current callset. |
| vat | Variant Annotation Table (VAT): per-site allele frequencies, ClinVar assertions, VEP consequence, and gene symbols. Queryable via SQL. |

BigQuery's columnar execution means queries touching only a few columns (position, filter label, gene symbol) scan only those columns, reducing per-query costs. GVS also includes a workflow for generating an associated Variant Annotation Table (VAT), which has functional annotations associated with the variants in GVS (e.g., gene symbols, population allele frequencies, ClinVar significance, and functional consequence). This enables SQL queries based on variant metadata.

## 2.2 Incremental Ingestion and Quality Filtering with VETS

GVS solves the N+k problem by treating ingestion as an append operation: each new sample's gVCF is converted to sparse alt-allele representation and written to vet_* and ref_ranges_* shards without updating existing rows. In other words, GVS detects samples that have already been loaded and does not load them again while new samples are loaded.

After ingestion, the variant filter set is updated with GATK VETS (Variant Extract-Train-Score) [5], an isolation-forest outlier detection model. VETS then applies variant-level filtering using allele-specific annotations: AS_QD, AS_MQRankSum, AS_ReadPosRankSum, AS_FS, AS_SOR, and AS_MQ (SNPs only). When new samples are ingested, VETS is retrained from scratch on the full updated callset rather than updated incrementally. An incremental approach was considered but determined to be unnecessary because VETS model training accounts for approximately 15% of total pipeline cost (see Table 2), making full retraining cost-justified and operationally simple. VETS is fully shardable and memory-efficient and runs 13× faster than VQSR. Retraining on CDR v9's 535,662 short read whole genome sequencing samples represented ~15% of total pipeline compute cost; *All of Us* retrains from scratch each CDR to maximize accuracy. CDR v9 VETS achieved SNP sensitivity ≈ 0.985, precision ≈ 0.999, and Indel sensitivity ≈ 0.971–0.990, precision ≈ 0.997–0.999 [1]. Of 1.38 billion total variant sites in *All of Us*, 71 million (5.1%) were soft-filtered, leaving 1.31 billion passing all filtering.

## 2.3 Multi-Format Extraction

GVS currently extracts to analysis-ready formats including VCF, Hail VDS (CDR v9 entire SNP/Indel callset: 37 TB), and PLINK pgen. Adopting a new format requires only a new extraction job, not callset reprocessing—the property that makes GVS a platform rather than a pipeline. Additionally, GVS includes pipelines for extracting these formats by genomic region and/or specified samples.

## 3 PRODUCTION DEPLOYMENT: *ALL OF US* CDR V9

CDR v9 ran ~3× faster than CDR v8 despite adding ~120,000 samples, made possible by building on top of data from CDR v8 rather than ingesting the entire sample set again. In concept, this is leveraging our previously described solution to the N+k problem. Further optimizations made included decreasing the Dataproc cluster size, resulting in fewer cluster crashes during data processing.

### 3.1 Operations Cost

GVS's queryable structure enables rapid answers to ad-hoc operational questions, such as why a variant is being filtered, participant

**Table 2: GVS short-read WGS runtimes across *All of Us* CDR versions.**

| Samples | Variant Sites | Runtime | Compute Cost/Sample | CDR Version | Ingestion Type |
|---|---|---|---|---|---|
| 415,000 | 1.2 B | ~6 weeks | USD$0.067 | CDR v8 | Full ingestion |
| 535,000 | 1.3 B | ~12 days | USD$0.035 | CDR v9 | N+k (partial ingestion) |

counts with specific variants (a common quality control (QC) question), or locating the origin of data errors. For example, when a program investigator reported that a common variant was not in the release, a SQL query against GVS spanning 414,000 samples and 1.5 trillion alternate genotypes returned the answer in one second at a cost of USD$0.01—immediately identifying the issue as upstream of GVS. More broadly, "how big is this problem?" questions (about variant frequencies, filter behavior across releases, or downstream tool failures) are routinely answered through direct SQL queries without re-running any part of the pipeline or spinning up analysis environments. These diagnostic queries often take under one second. Additionally, simple questions about the prevalence of variants in the data (and associated participants) can be answered with GVS and do not require the initialization of compute environments. Table 3 illustrates representative query times for pulling cohorts (not counts) of samples (without caching).

## 4 DOWNSTREAM DATA PRODUCTS ENABLED BY GVS

Because GVS is a single, authoritative, incrementally-updated source of truth, multiple services derive their inputs from it without redundant storage or reprocessing. We describe two examples below.

### 4.1 *All of Us* Cohort Builder

The *All of Us* Cohort Builder allows researchers to define cohorts using combined phenotypic and genotypic criteria. Genotypic queries are powered by GVS: the VAT is queried to identify participants carrying variants of interest (by functional annotations, such as gene symbol), with cached data transforms enabling sub-minute response times. The GVS extraction workflow is then called to create a whole genome VCF of the participants meeting both phenotype and genotype criteria for about USD$80 (*N*=5k of 414k).

### 4.2 *All of Us* + AnVIL Imputation Service

Imputation quality depends on reference panel size and diversity; panels dominated by European ancestry individuals limit accuracy in underrepresented populations. The *All of Us* + AnVIL Array Imputation Service [2] provides cloud-native imputation backed by the world's largest and most diverse reference panel: >515,000 whole genomes (414,830 AoU CDR v8 + 100,749 CCDG on AnVIL) phased with Beagle [4], covering 665 million high-quality sites (49% of the samples are genetically similar to non-European reference populations). The panel was built in part from GVS: CDR v8 VCFs extracted from GVS were merged with the AnVIL dataset after quality control and filtering. GVS's ability to produce consistent, reproducible extractions across hundreds of thousands of samples eliminates the need for a separate, redundant workflow and/or callset copies. The service runs on Terra (a secure, FedRAMP/NIST 800-53 Rev 5 Moderate environment) in the backend.

## 5 CODE AVAILABILITY

GVS is open-source, implemented as WDL workflows and Java and Python code within GATK: https://github.com/broadinstitute/gatk/tree/ah_var_store [3].

## 6 FUTURE DIRECTIONS

CDR v9 (535,662 samples, released June 26, 2026) continues GVS's trajectory toward the one-million-participant goal. Upcoming work in 2027 includes expanding extraction formats and releasing a GVS-based joint calling service which will enable external research teams to run production-grade joint calling on their own cohorts and extract to a supported analysis-ready format. On the imputation side, the *All of Us* + AnVIL lpWGS Imputation Service launched in August 2026; a Structural Variant (SV) Imputation Service is planned for 2026, using a long-read reference panel distinct from the array imputation panel.

## 7 RELEVANCE TO THE BIODMS COMMUNITY

GVS addresses core biomedical data management challenges at scale: cloud-native genomic data modeling, cost-effective petabyte-scale variant storage, scalable quality filtering without full reprocessing, and interactive querying over trillion-observation genotype matrices. The *All of Us* Cohort Builder and the *All of Us* + AnVIL Imputation Services demonstrate how a well-designed database storage layer naturally enables downstream operations, analyses, applications, and services that would otherwise require redundant infrastructure or very specialized skillsets.

**Table 3: Example queries over 414,830 CDR v8 samples.**

| Query | Time to Return Result | Joined with VAT |
|---|---|---|
| All participants with chr11:6868417-C-A; a known cilantro-taste variant | 6 seconds | No |
| All participants with a stop-gained variant in BRCA1 | 56 seconds | Yes |
| All participants with a ClinVar P/LP variant(s) in any of the fourteen MODY genes with GQ>20 and total AD >20 at that site | 67 seconds | Yes |

# 8 CONCLUSION

The Genomic Variant Store decouples cloud-native storage from analysis formats to enable population-scale genomic services. Deployed on 535,662 *All of Us* WGS samples, GVS delivers sub-minute interactive queries, cost-effective incremental ingestion, and reproducible multi-format extractions. This shared storage layer reduces operations costs and maintenance burden. The database itself enables both the *All of Us* Cohort Builder and the generation of the world's most diverse imputation reference panel—demonstrating that decoupled, cloud-native, extensible variant storage is a model for biobank-scale genomic data management.

## ACKNOWLEDGMENTS

The authors thank *All of Us* Research Program participants, the NHGRI, and the AnVIL team. This work was made possible by National Institutes of Health (NIH) awards: (1) OT2OD035404, "All of Us Data and Research Center (DRC);" (2) OT2OD03821, "Broad-Color: The Genome Center for the Future of All of Us;" (3) OT2OD002750, "The Broad-LMM-Color Genome Center for All of Us," funded by the NIH Office of the Director; and (4) U24HG010262, "AnVIL: A National Resource for Genomic Data Analysis and Visualization," funded by the National Human Genome Research Institute.

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

## AUTHORS

**M. Morgan Aster** is a Principal Software Engineer at the Broad Institute's Scalable Services Platform, developing cloud-native infrastructure for the *All of Us* + AnVIL Imputation Service. She earned her PhD in Neuroscience from UPenn and a BA from Yale.

**M. Kate Balaconis** is a Program Manager in the Broad Institute's Data Sciences Platform, serving as project manager for AnVIL, BioData Catalyst, and related cloud infrastructure projects. She holds a PhD from Northeastern University.

**Matt Bemis** is a Principal Software Engineer at the Broad Institute driving end-to-end engineering for the *All of Us* + AnVIL Imputation Service. He holds a BS in Computer Science from UConn.

**Miguel Covarrubias** is a Principal Software Engineer on the Variants team at the Broad Institute, developing cloud-native genomic data infrastructure for GVS across 500,000+ samples on Google BigQuery and Hail.

**Aaron Hatcher** is a Principal Software Engineer and technical lead of GVS at the Broad Institute, optimizing biobank-scale variant data storage, manipulation, and extraction for low per-sample cost on cloud platforms.

**Christopher Kachulis** is Director of Computational Science at the Broad Institute and Broad Clinical Labs, with significant expertise in imputation and polygenic risk scores. He holds a PhD in Physics from Boston University.

**Sofia Labrecque** is a Senior Product Manager at the Broad Institute managing GVS and genomics pipelines leveraging the *All of Us* + AnVIL reference panel. She holds a BA from Duke University and an MPH from Harvard University.

**Saloni P. Shah** is a Senior Software Engineer at the Broad Institute's Scalable Services Platform, building and optimizing services for large-scale scientific workflows. She holds an MS from Northeastern University.

**Lee Lichtenstein** is Director of Applied Science at the Broad Institute, leading large-scale genomics data processing and production-grade scientific infrastructure for datasets including *All of Us*.
