# OpenReview forum: "The Genomic Variant Store: Decoupling Cloud-Native Storage from Analysis to Power Population-Scale Genomic Services"
_VLDB.org/2026/Workshop/BioDMS — BioDMS 2026 ProjectTalk_

### Official Review · Reviewer_bYdK · 2026-06-10

**Summary:**

The paper presents the Genomic Variant Store (GVS), a cloud-native data management system built on Google BigQuery for population-scale joint variant calling.
The core idea is to employ a sparse alt-allele storage with reference-block modeling, enabling linear storage scaling, incremental ingestion,  ML-based filtering via GATK VETS, and interactive SQL-based variant search.
GVS is the production backend for the All of Us program, and the paper offers it as a case study for the BioDMS community.

**Confidence Of Review:**

3

**Detailed Feedback Points:**

1. Real-world, production-grade scale system case study. This will immensely help the data management community to learn about large-scale DB challenges in genomics world.
2.  The "nail/hammer" structure cleanly motivates each design decision against a specific legacy failure mode.

**Relevance For Biodms:**

4

---

### Official Review · Reviewer_3SnX · 2026-06-16

**Summary:**

The submission presents the Genomic Variant Store and frames it as a data management engine for genomic variant data. One of the main parts of motivation is joint variant calling, leveraging dataset at the scale of hundreds of thousands of genomes (with 1 million being a feasible target). GVS aims to support large scale analyses using Google BigQuery to improve handling of the large data volumes and avoid costly re-analyzes once new samples enter the dataset.

**Confidence Of Review:**

3

**Detailed Feedback Points:**

Strong points:
- GVS is already more or less a ready product, with further development potential. Might be an interesting example of the real-world convergence of biomedical challenges and data management solutions
- Relevant type of data, complex data management challenge, good motivation, very useful to the genomics community

Neutral:
- focuses on both a hammer and a nail, which is technically not fully "ideal" according to the call for contributions, but here seems to really work nice and could be a good discussion starter for the emerging community

Opportunities for improvement:
- it would be good to motivate more quantitatively why and how much does joint calling improve precision - how large is the performance improvement compared to traditional variant calling, that it motivates this scale of data infrastructure? This might not be entirely clear from the current write-up
- please make the code publicly available (incl. a link in the manuscript)

**Relevance For Biodms:**

3

---

### Official Review · Reviewer_cEbu · 2026-06-20

**Summary:**

This paper describes the GVS, a production system from the Broad Institute that manages a large callset of the "All of Us" program. The results are interesting, and the paper offers a valuable real-world perspective for the BioDMS communit

**Confidence Of Review:**

3

**Detailed Feedback Points:**

- I suggest adding more technical details about the evaluation. There is a lack of information provided about the table schema, typical queries and their performance, the details of incremental ingestion, or concurrency management.
- Although execution times are provided, a critical analysis of the performance is missing.
- Why did CDR v9 (535k genomes) take only 12 days, while CDR v8 (415k genomes) took about 6 weeks? This substantial difference is not explained.
- Furthermore, no quality metrics for the callset compared to other methods are presented, which is fundamental for a system of this type.
- Could you elaborate on the mechanics of incremental ingestion? In particular, how do the authors handle updating the filtering model (VETS) when new samples are added?
- Is the model retrained from scratch, or is there an incremental update procedure?
- The difference in execution time between different releases is evident: from 6 weeks to 12 days for an increase of 120k genomes. Can the authors provide an explanation for this?

**Relevance For Biodms:**

3